# Evolution of Annealing Twins in a Hot Deformed Nickel-Based Superalloy

**DOI:** 10.3390/ma15010007

**Published:** 2021-12-21

**Authors:** Yu-Chi Xia, Xiao-Min Chen, Yong-Cheng Lin, Xian-Zheng Lu

**Affiliations:** 1School of Mechanical and Electrical Engineering, Central South University, Changsha 410083, China; csuimcm@163.com; 2College of Automotive and Mechanical Engineering, Changsha University of Science and Technology, Changsha 410114, China; xzlucsu@163.com

**Keywords:** annealing twins, superalloy, hot deformation, dynamic recrystallization, grain boundary

## Abstract

The hot deformation characteristics of a GH4169 superalloy are investigated at the temperature and strain rate ranges of 1193–1313 K and 0.01–1 s^−1^, respectively, through Gleeble-3500 simulator. The hot deformed microstructures are analyzed by optical microscopy (OM), transmission electron microscopy (TEM), and electron backscattered diffraction (EBSD) technology. The effects of deformation parameters on the features of flow curves and annealing twins are discussed in detail. It is found that the shapes of flow curves are greatly affected by the deformation temperature. Broad peaks appear at low deformation temperatures or high strain rates. In addition, the evolution of annealing twins is significantly sensitive to the deformation degree, temperature, and strain rate. The fraction of annealing twins first decreases and then rises with the added deformation degree. This is because the initial annealing twin characters disappear at the relatively small strains, while the annealing twins rapidly generate with the growth of dynamic recrystallized grains during the subsequent hot deformation. The fraction of annealing twins is relatively high when the deformation temperature is high or the strain rate is low. In addition, the important role of annealing twins on dynamic recrystallization (DRX) behaviors are elucidated. The obvious bulging at initial twin boundaries, and the coherency of annealing twin boundaries with dynamic recrystallized grain boundaries, indicates that annealing twins can motivate the DRX nucleation during the hot deformation.

## 1. Introduction

For polycrystalline materials, the grain boundary characteristics are notably significant and exceedingly affect both the mechanical and physical properties of a material [1,2]. For example, the diffusion/sliding/precipitation behaviors of grain boundary significantly depend on the grain boundary features [3]. Grain boundary engineering (GBE) is an effective method to optimize the grain boundary features for improving material properties [4]. Such improvements result from the increased “special boundaries,” which occupy many coincident sites lattice (CSL) adjacent grain boundaries [5,6,7]. The so-called “special boundaries” usually refer to $\sum 3^{n}$ (*n* = 1, 2, 3) twin boundaries. Previous studies show that a high concentration of $\sum 3^{n}$ twin boundaries can improve material properties [8,9]. Recently, many studies have been involved in the evolution of boundaries in various metals and alloys during hot working. According to the formation mechanism of $\sum 3^{n}$ boundaries, there are two kinds of twinning, i.e., deformation twinning and annealing twinning. Generally, deformation twinning is induced by dislocation substructures during plastic deformation [10]. Khoddam et al. [11] found that “heterogeneous deformation twinning” can cause significant surface alteration during the uniaxial tensile straining of a TWIP steel. It is generally accepted that the formation of annealing twin is mainly associated with grain boundary migration during the static or dynamic recrystallization [12,13]. Mirzadeh et al. [14] found that lots of annealing twins generate after the initiation of dynamic recrystallization (DRX) in a hot deformed austenitic stainless steel. Cao et al. [15] found that annealing twin boundaries increase with the volume fraction of DRX of a nickel-chromium alloy.

Because of its excellent high-temperature strength, creep, and corrosion resistance, nickel-based superalloy is extensively applied in the “hot section” critical parts of modern aircraft engines, such as turbine disks [16,17,18]. During the last few decades, many researchers have studied the thermal deformation behavior of nickel-based superalloys [19,20,21,22,23,24,25,26,27,28,29,30,31,32,33]. Zhang et al. [19], Wen et al. [20,21], and Ning et al. [22], found that δ phase greatly affects the DRX and flow behaviors of GH4169 superalloy during hot deformation. However, too many δ phases can lead to flow instability. Chen et al. [23] established a segmented DRX kinetic model for GH4169 superalloy. Liu et al. [24] investigated the DRX behaviors of GH4169 superalloy by cellular automation method. Shore et al. [25] found DRX is the reason for the improvement of ductility of Incoloy 901 alloy at high temperatures. Li et al. [26] and Wang et al. [27] found the annealing twins generate during the DRX of hot deformed nickel-based superalloys and can be an assistant nucleation mechanism for DRX. Chong et al. [28] enhanced the microstructure of Inconel 740 H superalloy designed for the ultra-supercritical coal-fired power plants via grain boundary engineering. In order to determine the safe and stable hot working domains, Wen et al. [29], Liu et al. [30], Shen et al. [31], and Zhang et al. [32,33] identified the optimal hot deformation parameters of some typical Ni-based superalloys by processing maps.

Although some efforts have been invested in the thermal deformation features of different nickel-based superalloys, the systematic investigations on twin-related grain boundary engineering are extremely limited. To better understand the annealing twin charterers and their effects on DRX, it is necessary to further understand the evolution of annealing twins during hot deformation. This study aims to elucidate the characters of the annealing twin and its contribution to the nucleation and growth of DRX grains. Firstly, the evolution of annealing twin is quantitatively measured. Then, the effects of strain, deformation temperature, and strain rate on the fraction of annealing twins are discussed. Finally, the role of annealing twins on DRX behaviors is investigated.

## 2. Materials and Experimental Procedure

The experimental material is forged GH4169 superalloy. Its primary chemical compositions (wt. %) are given in Table 1 [34]. Cylindrical specimens of 8 mm in diameter and 12 mm in height were cut from the hot forged bar. To obtain the uniform initial microstructure, the samples were solution treated at 1313 K held for 45 min, and rapidly cooled by water. Using a Gleeble-3500 equipment, several hot compression experiments were carried out under 1193–1313 K and 0.001–1 s^−1^. The experimental deformation temperature and strain rate is selected according the suitable hot forming domain of the tested superalloy [29]. In order to reduce the frictions during hot deformation, the tantalum foil with a thickness of 0.1 mm was placed between the specimen and dies. Before compression, the samples were soaked to the planned deformation temperature at a heating rate of 10 K/s and kept as long as 300 s to eliminate the thermal gradients. Subsequently, the samples were deformed to the initial height reduction of 12–70%, and rapidly cooled by spraying water.

Optical microscopy (OM) (Olympus Corporation, Tokyo, Japan), Transmission Electron Microscopy (TEM) (Tecnai G2 F20; FEI company; Hillsboro, OR, USA) and electron backscatter diffraction (EBSD) (FEI Electron Optics B.V; Prague, Czech Republic) technologies were employed to analyze the deformed microstructures. Samples for OM analysis were cut down along the compression axis and mechanically polished, followed by chemical etching. Chemical etching was carried out in a solution consisting of HCl (100 mL) + CH_3_CH_2_OH (100 mL) + 5 gCuCl_2_. The foils for TEM and EBSD were ground to 70–80 μm and thinned by twin-jet electropolishing using a perchlorate alcohol solution (1:9 in volume). The orientation data were analyzed using HKL Channel 5 software. The DRX volume fraction and annealing twin fraction can be directly measured from the orientation data of EBSD analysis.

## 3. Results and Discussion

### 3.1. Initial Microstructure

Figure 1a illustrates the solution treated microstructure of the studied superalloy. The equiaxed grains and annealing twins with lamella-like straight features are occupied the initial microstructure. The initial annealing twins are induced by the intergranular thermal residual stresses [35]. Figure 1b shows the grain boundary distribution map. According to the ‘coincidence site lattice’ (CLS) model [36], which is widely used to characterize grain boundaries, the grain boundaries can be divided into three categories: low angle grain boundaries (LAGBs, misorientation angle small than 15°), low-∑ CLS boundaries (3≤∑≤29), and general high angle grain boundaries (HAGBs, misorientation angle higher than 15°) [15]. Here,∑ is named as the reciprocal density of coincident sites. For example, a ∑3 boundary means 1/3 of the atoms in coincident sites [37]. In Figure 1b, the LAGBs, HAGBs, and low-∑ CLS boundaries are marked with thin-gray, thick-black, and thick-blue lines, respectively. Figure 1c shows the distribution of misorientation angle. In Figure 1c, there are pronounced peaks at about 60° and 38.9°. It indicated that the initial grain boundaries consist of the first-order ∑3 boundaries characterized by 60°<111> misorientation, as well as the second-order ∑9 boundaries characterized by 38.9°<110> misorientation. The fractions of ∑3 and ∑9 boundaries are evaluated as 52.71% and 2.43%, respectively. Obviously, the number of ∑9 boundaries is extremely less than that of ∑3 boundaries. Moreover, due to the low grain boundary energy, the coherent twin boundaries with ∑3 boundaries possess desirable features in GBE materials [1]. Therefore, the evolution of ∑3 boundaries is mainly discussed in this study. Here, ∑3 boundaries are synonymous to annealing twins [37,38].

### 3.2. Hot Deformation Behavior

Figure 2 displays the flow stress during the hot compression of the studied superalloy. In Figure 2a, the flow curves exhibit the pronounced peak stress followed by a steady stage at relatively high deformation temperatures (*T* = 1253 and 1313 K), indicating the occurrence of work hardening (WH), dynamic recovery (DRV), and DRX. At relatively lower deformation temperatures (*T* = 1193 and 1223 K), the peak stress becomes indistinct and exhibits a plateau. This is because the low temperature diminishes the grain boundary mobility, and delays the growth of DRX grains [39]. Moreover, the DRV induced by dislocation climbing and cross slip is intensive at low temperatures. Thus, it needs a long period to achieve the dynamic balance between the work hardening and dynamic softening. Accordingly, an obvious stress plateau occurs. In addition, the flow stress is also sensitively influenced by strain rate. For a given deformation temperature (i.e., 1193 K), the flow stress curves exhibit a broad peak when the strain rate is low. Nevertheless, the broad peak disappears at fairly high strain rates (0.1 and 1$s^{-1}$), and the continuous flow softening occurs. Similar results were obtained in other nickel-based superalloys, such as Incoloy 901 [40] and Allvac 718Plus [41]. Momeni et al. [40] and Mitsche [41] have reported that the rise of strain rate can result in the increased DRX grains and their sizes; thus, there is a continuous flow softening as the strain rate is relatively high [42,43,44,45]. Nonetheless, the DRX degree decreases with the raised strain rate [46,47,48]. It is because that the reduction of deformation time at a high strain rate limits the growth of DRX grains. [49,50]. The elongated grain boundaries can confirm the absence of DRX behavior when the strain rate is relatively high (Figure 3). Consequently, the continuous flow softening at a comparatively high strain rate is mainly induced by the adiabatic heating [15], i.e., the deformation at relatively high strain rates easily results in a swift temperature rise in material with the further straining [51].

Figure 2b illustrates the variations of peak stress and yield stress under the tested conditions. It is visible that both the peak stress and yield stress are also dependent on the deformation parameters. The peaks and yield stress increases with the increased strain rate or the decreased strain rate. Moreover, it is interesting that the difference between the peak and yield stress is large at relatively low temperatures or high strain rates. It is well known that peak stress represents the first balance due to work hardening and dynamic softening, while the yield stress means the start of plastic deformation. The large difference between the peak and yield stress results from the slow dynamic softening. Because the low deformation temperature restrains the thermal activity and the high strain rate decreases the deformation time, it is difficult for the dislocation annihilation induced by DRX and DRV [52].

### 3.3. Dynamic Recrystallization Behavior

Figure 4a–e depicts the impacts of strain on hot deformed microstructure at 1223 K and 0.01 s^−1^. The corresponding strains are illustrated in Figure 4f. In Figure 4f, the peak strain, which corresponds to the peak stress, is 0.27 (marked with point ‘B’). The strain of 0.12 (marked with ‘A’) is smaller than the critical strain for the initiation of DRX [23]. The strains of 0.52, 0.8, and 1.2 (marked with ‘C’, ‘D’, and ‘E’, respectively) depict the different deformation levels larger than the vertex strain. In Figure 4a, the grain boundaries are straight, and there are almost no DRX grains when the true strain is 0.12. It is interesting that DRX nucleation predominantly occurs in the regions with high strain gradients, such as grain boundaries and triple points (Figure 4b). With the further straining, the nucleation of DRX grains by strain-induced grain boundary bulging (Figure 4c,d) can be found. In some regions marked with rectangle zones in Figure 4c,d, DRX grain boundaries are covered by annealing twins. This indicates that new annealing twins form during the growth of DRX grains. In Figure 4e, it is observed that the deformed matrix is mainly consumed by DRX grains at relatively large strains.

### 3.4. Effects of Deformation Parameters on the Evolution of Annealing Twins

#### 3.4.1. Effects of Strain

Figure 4 also summarizes the influences of strain on the features of annealing twins. Compared with the initial microstructure before hot deformation, the clear alterations of annealing twin can be found in the hot deformed material. Figure 4a shows the annealing twins at a relatively small strain (0.12). It can be found that that the annealing twins are mainly visible at the original grain boundaries, and the number of annealing twins rapidly decreases at the initial deformation stage. This is because that hot deformation significantly affects the characteristics of annealing twins [53]. It is well known that progressive deformation and corresponding grain rotation can lead to the rapid loss of original twin characters [54]. Especially, a majority of ∑3 boundaries lose their ideal misorientation (i.e., 60°<111>) and become ordinary HAGBs. In Figure 4a, some annealing twin boundaries are severely distorted, while others exhibit minimal distortions. This may be related to the incompatibility of deformation among different grains. With the increase of strain, the number of lamella-like straight annealing twins further decreases. Meanwhile, the newly-formed twins appear at DRX grain boundaries or within DRX gains, as shown in Figure 4b. Usually, no annealing twins form behind the grain boundary bulging region during the nucleation of DRX [54]. The generation of annealing twins may be attributed to the DRX mechanism. According to the authors’ previous studies [39], the nucleation of DRX is mainly characterized by grain boundary bulging, which often refers to strain-induced boundary migration.

Meanwhile, it can be found the nucleation of DRX mainly along the primary grain boundaries, as shown in Figure 4c,d. With the further deformation, some DRX grains undergo the deformation. The driving power for the evolution of DRX grain will be reduced with the increased dislocation density inside the deformed DRX grains [55]. Therefore, the growth of DRX grain may be stopped, i.e., a growth accident occurs. Due to the occurrence of growth accident, the stacking sequence of the atomic layer at the {111} plane grain corner may accidentally fault, and, thus, the annealing twins form near grain boundaries. In other words, the annealing twins generate by grain boundary migration rather than slip activity within grains [56]. Wang et al. [57] reported that the annealing twinning associates with recrystallization rapidly reduces the total stored deformation energy in the deformed material. Interestingly, the evolution of total annealing twins shows different trends with the increased strain (Figure 4c,d). It reveals that the number of distorted twins is reduced (Figure 4c), while the number of newly-formed short twins at the DRX grain boundaries increases (Figure 4d). At the strain of 1.2, most grain boundaries are covered by the new-formed annealing twins, as shown in Figure 4e.

Figure 5a illustrates the impacts of strain on misorientation distribution at 1223 K and 0.01 $s^{-1}$. In Figure 5a, the misorientation angle peak mainly appears at about 60°, which indicates the appearance of ∑3 annealing twins. For the solution-treated material, the misorientation angle peak is relatively intense and concentrated, which indicates that the initial annealing twins have ideal misorientations. However, the misorientation angle peak becomes significantly weakened and broad with the increase of strain. This is because the initial annealing twins remarkably deviate from their ideal misorientations. Figure 5b represents the variations of annealing twin fraction and DRX volume fraction with strain. At the strain of 0.12, the DRX volume fraction is close to zero, while the annealing twin fraction rapidly drops from 52.71% (solution-treated) to 13.72%. With the further straining, the DRX volume fraction rapidly increases. However, the fraction of twin boundaries firstly decreases and then gradually increases with the increased strain. This is because the loss and generation of annealing twins are competitive processes. If the growth rate of new twins is lower than the loss rate, the annealing twin fraction will decrease. At the strain of 0.52, the fraction of twin boundaries reaches 3.79%, which is the minimum value under the tested condition. When the strain is higher than 0.52, it is obvious that the number of annealing twins become large when the DRX volume fraction is multiplied, as depicted in Figure 5b. As reported by Pande et al. [58], the number of annealing twins depends on the driven force of grain boundary migration. More precisely, the increment of the number of annealing twins is proportional to the increment of DRX grain size. If the nucleation of DRX grain takes place, the DRX nuclei will step by step grow up with the further straining till a stable state of DRX is reached. Thus, the fraction of annealing twins gradually increases at relatively high strain.

#### 3.4.2. Effects of Deformation Temperature

Figure 4e and Figure 6 illustrate the evolution of annealing twins with deformation temperature at the strain and strain rate of 1.2 and 0.01 s^−1^, respectively. There are only a few annealing twins at the DRX grain boundaries at 1193 K and 1253 K (Figure 4e and Figure 6a). The number of annealing twins obviously increases at 1253 K (Figure 6b). In addition, the annealing twins become more and more prevalent at DRX grain boundaries. Figure 7a depicts the misorientation angle distribution at deformation temperatures. In Figure 7a, it is observed that the peak corresponding to ∑3 boundaries is importantly dependent on the deformation temperature. The intensity of peaks increases progressively as the deformation temperature is increased, indicating the progressive increase of annealing twin boundaries. Meanwhile, in Figure 7b, the fraction of annealing twin significantly increases when the DRX volume fraction increases. A similar phenomenon can be observed in other alloys, such as Ti-modified austenitic stainless steel [59] and nickel-chromium alloy (800 H) [15]. As mentioned in Section 3.4.1, the increased number of annealing twins is tightly associated with the growth of DRX grains. Since the DRX is the thermal activation process related to atomic diffusion, the high deformation temperature promotes the average kinetic energy of atoms, as well as the grain boundary mobility. Thus, the DRX grains size increases with the increased deformation temperature, which accelerates the formation of annealing twins during hot deformation. Accordingly, the fraction of annealing twins markedly increases when the deformation temperature is considerably high.

#### 3.4.3. Effects of Strain Rate

Figure 6b and Figure 8 show the influences of strain rate on the evolution of annealing twins [39]. Here, the true strain is 1.2, and the deformation temperature is 1253 K. The coherency of annealing twin boundaries and DRX grain boundaries can be found after hot deformation, as shown in Figure 6b and Figure 8. To understand the evolution of annealing twins at various strain rates, the misorientation angle distribution and the variations of annealing twin fraction with strain rate are calculated, as shown in Figure 9. In Figure 9a, the misorientation angle peak declines with the raised strain rate, which implies that the evolution of annealing twin boundaries is greatly influenced by the strain rate. The fractions of annealing twins can be determined as 8.96%, 8.06%, and 6.57% when the strain rates are 0.001, 0.01, and 0.1 s^−1^, respectively. In other words, the fraction of annealing twins is declined at a fairly high strain rate. The decrease of annealing twin boundary fraction with strain rate can be explained by the influences of strain rate on the growth of DRX grain. When the strain rate is relatively high, the growth of DRX grains is strictly restricted by the limited deformation time. In other words, the grain boundaries migration is restrained at a relatively high strain rate. Thus, the fraction of annealing twins is reduced by the increased strain rate. Figure 9b shows the variations of annealing twin fraction and DRX volume fraction with strain rate. In Figure 9b, the variation of annealing twin fraction is associated with the DRX behavior of the studied superalloy. This is because a high strain rate decreases the DRX volume fraction, and the DRX grains size. So, the fraction of annealing twin is small at a relatively high strain rate.

### 3.5. The Role of Annealing Twins on DRX

As has been discussed in Section 3.4, the annealing twins include the initial and newly-formed annealing twins during hot deformation. In this section, the role of initial and newly-formed annealing twins on DRX will be discussed.

#### 3.5.1. The Role of Initial Annealing Twins

As discussed in Section 3.4.1, the initial annealing twins easily lose their character due to the rotation of grains during hot deformation. DRX nuclei can be bulged from the rotated initial annealing twins, as shown in Figure 10a,b. This is because the annealing twins can contribute to strain hardening and convert active barriers for slip. Consequently, the extensive initial twinning can result in the accumulated strain energy in the deformed superalloy, which in turn facilitates the bulging of local grain boundary [35]. Figure 11 shows TEM morphology of the annealing twins at 1253 K and 0.001 $s^{-1}$ (the strain is 0.27). In Figure 11, the dislocations cannot pass through the annealing twin boundaries, leading to the high dislocation gradient at the local annealing twin boundaries. This is helpful for bulging nucleation. Favre et al. [60] and Mirzadeh et al. [14] have also reported that it is easy for the new DRX nuclei to bulge from the twin boundaries. In addition, in Figure 10c, it can also be found that the initial twin boundaries can become the DRX boundaries at the early deformation stage. This is because the twinning boundaries can be transformed into considerable mobile grain boundaries, and new HAGBs are directly formed. Therefore, the initial annealing twins can enhance the nucleation of DRX.

#### 3.5.2. The Role of Newly-Formed Annealing Twins

As shown in Figure 4, Figure 6, Figure 8 and Figure 10, it can be effortlessly discovered that fresh annealing twins emerge at DRX grain boundaries or the interior of DRX grains. Figure 12 shows the sketch map of DRX grains development [61]. For the studied superalloy, the basic nucleation mechanism is bulging nucleation induced by the migration of grain boundary, as revealed in Figure 12a. Usually, new grains can be separated by a bridging subgrain wall (Figure 12b). For the studied superalloy, it can be found that the annealing twins near the bulging interface can facilitate the separation of the bulged portions from the initial grains (Figure 12b). Therefore, it can be concluded that the newly-formed annealing twins can also play a vital role in DRX nucleation.

## 4. Conclusions

The hot deformation behavior of GH4169 superalloy is studied. The effects of deformation parameters on the features of annealing twins are investigated. Some important conclusions can be summarized as follows:(1)The peak stress becomes indistinct and exhibits a plateau at lower deformation temperatures and strain rate, showing typical work hardening, dynamic recovery, and dynamic recrystallization features. Because the dynamic softening at low deformation temperature or high strain rate is intense, a large difference between the peak and yield stresses can be observed.(2)The evolution of annealing twin is sensitively affected by deformation parameters. At the beginning of deformation, the initial annealing twins are easy to disappear due to the rotation of grains. New annealing twins form on the DRX grain boundaries or within DRX grains with the increased strain. It indicates the grain boundary migration during the growth of DRX grains can promote the formation of annealing twins. At high deformation temperatures or low strain rates, DRX is accelerated. Accordingly, the fractions of annealing twin boundaries significantly increases.(3)The annealing twins play a vital role in the nucleation of DRX. The initial annealing twins easily lose their character due to the rotation of grains during hot deformation, providing an extra driving force for the DRX nucleation. In addition, it can be transformed into considerable mobile grain boundaries of new DRX nucleation. Besides, fresh annealing twins, which emerge at DRX grain boundaries or the interior of DRX grains, can accelerate the separation of bulging nuclei from the initial grains.

## Figures and Tables

**Figure 1 materials-15-00007-f001:**
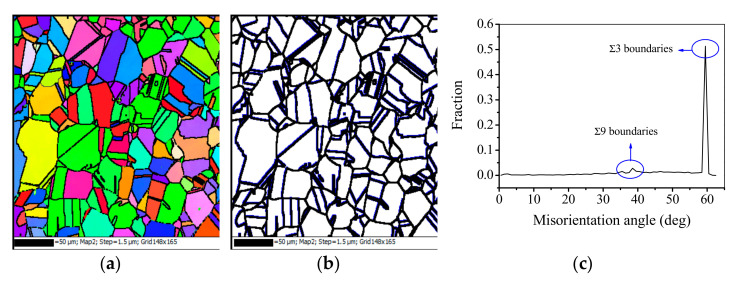
Microstructure of the studied superalloy before hot deformation: (**a**) orientation image microscopy map; (**b**) grain boundary distribution (LAGBs, HAGBs, and annealing twin boundaries are indicated by thin-gray, thick-black, and thick- blue lines, respectively); (**c**) misorientation angle distribution.

**Figure 2 materials-15-00007-f002:**
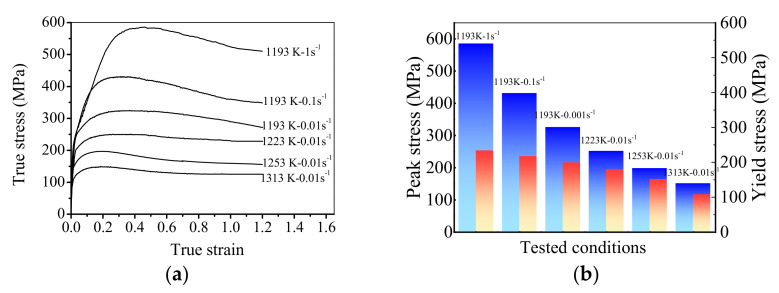
Flow stress of the studied superalloy during hot compression: (**a**) True stress-true strain curves; (**b**) variations of peak stress and yield stress under the tested conditions.

**Figure 3 materials-15-00007-f003:**
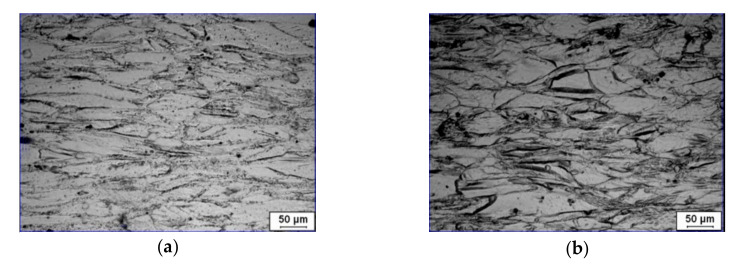
Optical micrographs of the deformed superalloy at the deformation temperature of 1193 K and strain rates of: (**a**) 0.1 s^−1^; (**b**) 1 s^−1^ (at the strain of 1.2).

**Figure 4 materials-15-00007-f004:**
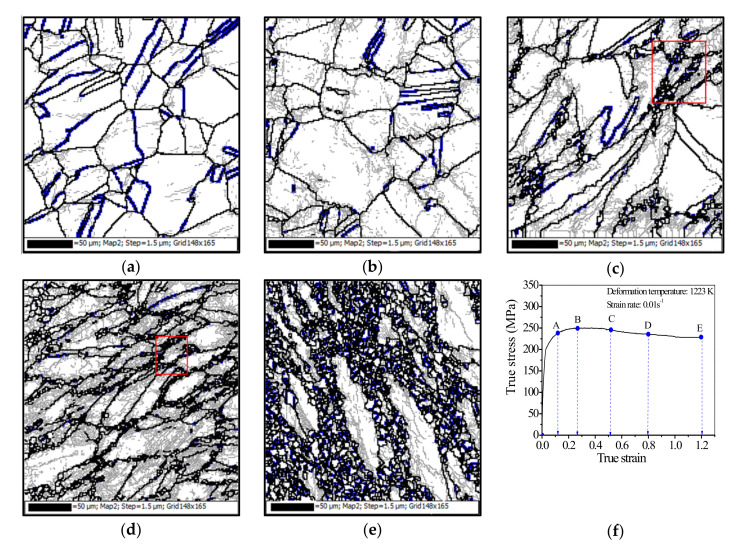
Orientation imaging microscopy maps at the strains of: (**a**) 0.12; (**b**) 0.27; (**c**) 0.52; (**d**) 0.8; (**e**) 1.2; (**f**) true stress-true strain curve (LAGBs, HAGBs and annealing twin boundaries are indicated by thin-gray, thick-black, and thick- blue lines, respectively).

**Figure 5 materials-15-00007-f005:**
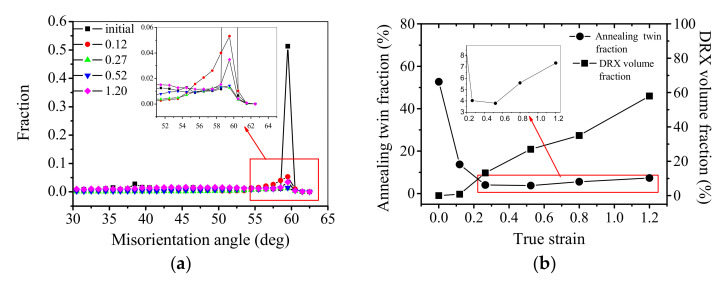
Changes of annealing twin fraction and DRX volume fraction at the deformation temperature of 1223 K and strain rate of 0.01 s^−1^: (**a**) misorientation angle distribution; (**b**) variations of annealing twin fraction and DRX volume fraction with strain.

**Figure 6 materials-15-00007-f006:**
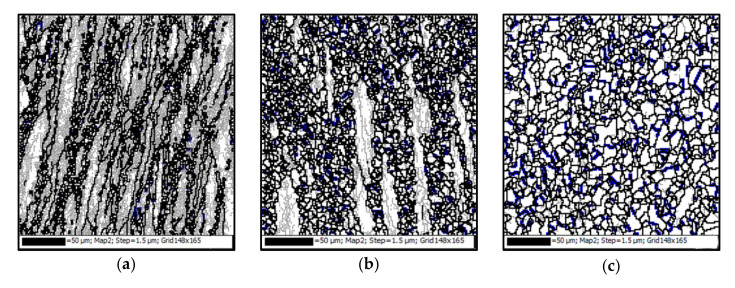
Evolution of annealing twins at the deformation temperatures of: (**a**) 1193 K; (**b**) 1253 K; (**c**) 1313 K (LAGBs, HAGBs and annealing twin boundaries are indicated by thin-gray, thick-black, and thick- blue lines, respectively. The strain and strain rate are 1.2 and 0.01 s^−1^, respectively).

**Figure 7 materials-15-00007-f007:**
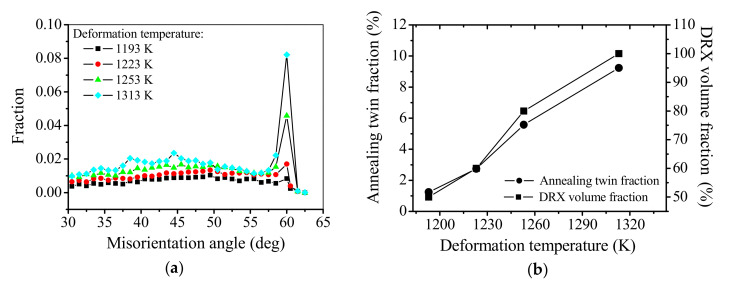
Changes of annealing twin fraction and DRX volume fraction at the strain and strain rate are 1.2 and 0.01 s^−1^, respectively. (**a**) misorientation angle distribution, and (**b**) variations of annealing twin fraction and DRX volume fraction with deformation temperature.

**Figure 8 materials-15-00007-f008:**
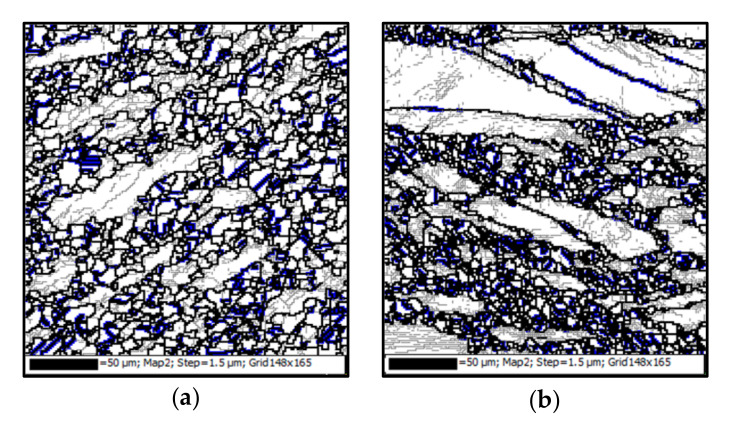
Evolution of annealing twins at the strain rates of: (**a**) 0.001 s^−1^; (**b**) 0.1 s^−1^ (LAGBs, HAGBs and annealing twin boundaries are indicated by thin-gray, thick-black, and thick- blue lines, respectively. The strain and the deformation temperature are 1.2 and 1253 K, respectively).

**Figure 9 materials-15-00007-f009:**
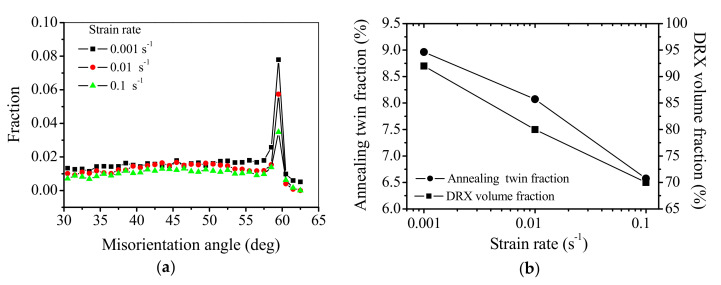
Changes of annealing twin fraction and DRX volume fraction at the strain and the deformation temperature are 1.2 and 1253 K, respectively. (**a**) Misorientation angle distribution and (**b**) variations of annealing twin fraction and DRX volume fraction with strain rate.

**Figure 10 materials-15-00007-f010:**
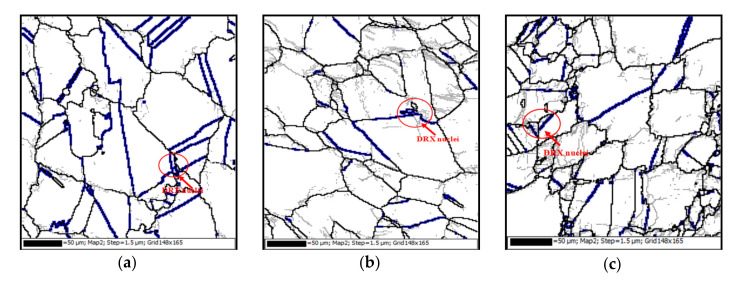
Orientation imaging microscopy maps at the strain of 0.27 and the deformation conditions of: (**a**) 1253 K, 0.001 $s^{-1}$; (**b**) 1253 K, 0.1 $s^{-1}$; (**c**) 1313 K, 0.01 $s^{-1}$ (LAGBs, HAGBs and annealing twin boundaries are indicated by thin-gray, thick-black, and thick- blue lines, respectively).

**Figure 11 materials-15-00007-f011:**
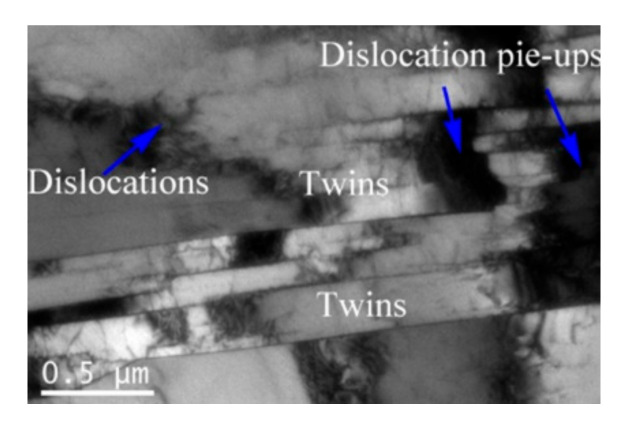
TEM morphology of the deformed superalloy at 1253 K, 0.01 $s^{-1}$ (The strain is 0.27.).

**Figure 12 materials-15-00007-f012:**
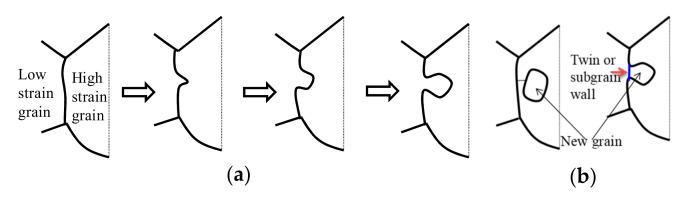
The development of DRX grains in the studied superalloy: (**a**) bulging nucleation due to grain boundary migration and (**b**) bulge separated from parent grain due to the formation of annealing twins or a bridging subgrains wall.

**Table 1 materials-15-00007-t001:** The primary chemical compositions (wt. %) of GH4169 superalloy.

Ni	Cr	Nb	Mo	Ti	Al	Co	C	Fe
52.82	18.96	5.23	3.01	1.00	0.59	0.01	0.03	Bal.

## Data Availability

The raw/processed data required to reproduce these findings cannot be shared at this time as the data also forms part of an ongoing study.

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
