# Peer review of "Evolution of Annealing Twins in a Hot Deformed Nickel-Based Superalloy"

_materials, 2021, doi:10.3390/ma15010007_

Round 1
Reviewer 1 Report
It is suggested to identify the commercial Ni-based alloy object of this study also through its commercial designation.
Author Response
Dear reviewer,
Thanks for the constructive advice from reviewers and editors. The manuscript has been carefully checked and improved again. The following are the responses to the comments from you. The corresponding modifications have been made and the important corrections are highlighted in the YELLOW color in the revised manuscript.
Point 1: It is suggested to identify the commercial Ni-based alloy object of this study also through its commercial designation.
Response 1: Thanks for the reviewer’s constructive comments. The experimental material is forged GH4169 superalloy. In the revised manuscript, the designation of the studied Ni-based alloy is given. The corresponding modifications have been made in SECTION ‘2. Materials and experimental procedure’ Page 2. Thanks so much!
Reviewer 2 Report
It’s a basic, but interesting research report concerning the influence of annealing twins on the evolution of the microstructure of nickel superalloy. The quality of the presented research results and discussions are at acceptable level. However, the following comments/suggestions should be taken into consideration by the authors:
- Why the authors deformed the samples to a total true strain in the range of 0.12-1.2? What was the reason for such strain range? Why not 0.1-1? This should be explained.
- What was the reason for selecting such strain rate range (0.001-1 s-1) for the compression tests? Such information should be introduced into the manuscript.
- What kind of lubrication the authors applied to minimize the influence of friction during Gleeble compression tests?
- How the authors measured the uniformity of the temperature distribution in the compression samples? The statement: “uniform thermal distribution” should be avoided in the case of Gleeble compression tests. Resistance heating always creates nonuniformity of the temperature distribution on the length of the compression sample – the authors should realize that. Keeping the compression samples at deformation temperature for 5 minutes prior to deformation does not influence the uniformity of the temperature distribution. It is always nonuniform due to the nature of resistance heating on Gleeble system.
Reviewer 3 Report
The presented article “Evolution of annealing twins in a hot deformed nickel-based superalloy” is interesting and written in good scientific language. It is worthy of being published in “Materials” magazine, but requires some corrections:
(1) It seems that the text in separate paragraphs is typed in a different font, for example, in “Abstract” the words “deformation”, “strain” are typed either in a not the same font, or slightly larger in size. In some places, there are no spaces, for example, in the “Initial microstructure” section in the last sentence of a paragraph. There are also some lexical errors, for example, repeating twice “On the one hand” page 6, first paragraph.
(2) It is necessary to correct the numbering of the links in accordance with the proposed magazine template
(3) DRX - decrypt at the moment of the first mention
(4) In “Materials and experimental procedure” put the diameter icon
(5) Figure 1, unfortunately, thin-gray, thick-black lines are hard to read, if it is possible to repaint in other colors, it would be great. But given that this will require changing all images, then it is worth evaluating this as a recommendation for the future.
(6) Correct the names of all figures: Figure instead of Fig, for example, in the second figure.
(7) Figure 2 would recommend making a slightly larger size, since this is one of the most significant graphs in this work. It would also be great next to the image in the table to give the values of the tensile strength, and if possible, the yield strength, depending on the temperature of deformation.
(8) Figure 5b in the enlarged fragment, check if there should be a square in the form of an icon, which is labeled as "DRX volume fraction"
Reviewer 4 Report
The manuscript presents a study about the hot deformation characteristics of a nickel-based superalloy by EBSD and hot compressive experiments. Also, the microstructure after deformation has been analyzed using optical microscopy and transmission electron microscopy. The paper needs minor revisions before it is processed further, some comments follow:
Abstract
There is no clear purpose of the study in the abstract and are not presented the characterizing methods. Also, please highlight the novelty of the study. The abstract must be reformulated.
Introduction section
The introduction section must be improved.
Please describe the acronyms in the manuscript text body where it was cited for the first time (e.g. DRX, EBSD).
The novelty of this study isn’t presented. Please include a paragraph, to highlight the novelty and the aim of this study.
Materials and experimental procedure
I think will be better if the chemical composition of the nickel-based superalloy should be presented in a Table, please improve.
Conclusion:
After such a long and quantitative result and discussion, the conclusion looks vague. Rewrite it.
